# Multidiscipline Stroke Post-Acute Care Transfer System: Propensity-Score-Based Comparison of Functional Status

**DOI:** 10.3390/jcm8081233

**Published:** 2019-08-16

**Authors:** Chung-Yuan Wang, Hong-Hsi Hsien, Kuo-Wei Hung, Hsiu-Fen Lin, Hung-Yi Chiou, Shu-Chuan Jennifer Yeh, Yu-Jo Yeh, Hon-Yi Shi

**Affiliations:** 1Department of Physical Medicine and Rehabilitation, Pingtung Christian Hospital, Pingtung 90059, Taiwan; 2Department of Internal Medicine, St. Joseph Hospital, Kaohsiung 80760, Taiwan; 3Division of Neurology, Department of internal medicine, Yuan’s General Hospital, Kaohsiung 80249, Taiwan; 4Department of Neurology, Kaohsiung Medical University Hospital, Kaohsiung 80756, Taiwan; 5School of Public Health, Taipei Medical University, Taipei 11031, Taiwan; 6Center for Neurotrauma and Neuroregeneration, Taipei Medical University, Taipei 11031, Taiwan; 7Department of Healthcare Administration and Medical Informatics, Kaohsiung Medical University, Kaohsiung 80708, Taiwan; 8Department of Business Management, National Sun Yat-sen University, Kaohsiung 80424, Taiwan; 9Department of Medical Research, Kaohsiung Medical University Hospital, Kaohsiung 80756, Taiwan; 10Department of Medical Research, China Medical University Hospital, China Medical University, Taichung 40447, Taiwan

**Keywords:** stroke, post-acute care, medical referral system, propensity score matching

## Abstract

Few studies have investigated the characteristics of stroke inpatients after post-acute care (PAC) rehabilitation, and few studies have applied propensity score matching (PSM) in a natural experimental design to examine the longitudinal impacts of a medical referral system on functional status. This study coupled a natural experimental design with PSM to assess the impact of a medical referral system in stroke patients and to examine the longitudinal effects of the system on functional status. The intervention was a hospital-based, function oriented, 12-week to 1-year rehabilitative PAC intervention for patients with cerebrovascular diseases. The average duration of PAC in the intra-hospital transfer group (31.52 days) was significantly shorter than that in the inter-hospital transfer group (37.1 days) (*p* < 0.001). The intra-hospital transfer group also had better functional outcomes. The training effect was larger in patients with moderate disability (Modified Rankin Scale, MRS = 3) and moderately severe disability (MRS = 4) compared to patients with slight disability (MRS = 2). Intensive post-stroke rehabilitative care delivered by per-diem payment is effective in terms of improving functional status. To construct a vertically integrated medical system, strengthening the qualified local hospitals with PAC wards, accelerating the inter-hospital transfer, and offering sufficient intensive rehabilitative PAC days are the most essential requirements.

## 1. Introduction

Acute stroke is a major cause of mortality and disability [1,2]. Stroke patients can incur considerable costs for medical care, including nursing, rehabilitative, and long term care. Therefore, many countries are attempting to establish comprehensive and integrated healthcare systems for stroke patients [3,4]. Post-acute care (PAC), which refers to medical care services that support the individual patient in recovery from illness or management of chronic disability, is aimed at enhancing the functional status of patients discharged from acute hospitalization [2,3,4,5]. Discharges to PAC facilities have increased nearly 50% during the past 15 years, and PAC is a major contributor to hospitalization costs in the United States [5]. To control medical expenses, reform the medical referral system, and improve continuity of care, the Taiwan National Health Insurance Administration (NHIA) focused on stroke for its first national PAC project in 2014—post-acute care for cerebrovascular disease (PAC-CVD).

In Taiwan, beneficiaries are free to visit their preferred physicians and are not required to follow strict referral rules [6]. An efficient PAC system for stroke patients is needed to reduce unnecessary utilization of hospital resources and to ensure seamless care for these patients [7]. Stroke patients and their families expect that local hospitals can deliver PAC at a lower cost and with greater efficiency, effectiveness, and convenience. Our review of studies published in international journals, however, shows that most studies of PAC have analyzed a limited number of patients in a single hospital [8,9]. Additionally, few studies have used longitudinal data exceeding 1 year, and few studies have applied propensity score matching (PSM) in a natural experimental design to examine the longitudinal impacts of a medical referral system on functional status. Therefore, this study coupled a natural experimental design with PSM to assess the impact of the medical referral system in stroke patients and to examine the longitudinal effects of the medical referral system on functional status.

## 2. Materials and Methods

### 2.1. The PAC Program

In Taiwan, the multidisciplinary PAC stroke team consisted of neurologists, physiatrists, physical therapists, occupational therapists, speech therapists, and nurses. The PAC rehabilitation program was prescribed by the physiatrist, and it consisted of a complex program of universal activities that were performed at least three times per day. One hour of physical therapy, occupational therapy, or speech and swallowing therapy was carried out at each time. Notably, the fiscal incentive for a medical center to transfer a patient to a regional or a district hospital is mitigated by several factors, including the PAC-CVD transfer policy, the willingness of the stroke patient (or family) to accept further post-acute care in local hospitals, and whether the physician agrees to the transfer. These factors should cause health providers to reconsider the manner in which patient stays are controlled and to be mindful that the shortest lengths of stay (LOS) may not obtain the best outcome for the patient [10,11].

### 2.2. Study Design and Sample

The study population included all stroke patients admitted to PAC wards in four Taiwan hospitals between March 2014 and March 2018 (defined as ICD-9-CM codes 433.x, 434.x, and 436.x for ischemic stroke, and codes 430 and 431 for hemorrhagic stroke). The inclusion criteria were acute stroke and admission to PAC ward within 40 days after day of stroke onset. Another inclusion criterion was Modified Rankin Scale (MRS) level 2 to 4. Instead of focusing on patients who had received intensive in-patient rehabilitation, this national PAC project focused on the prevention of complications (e.g., pressure sores) in stroke patients who were bed-ridden (MRS 5). Patients who did not have major disability (MRS 0–1) were assumed to have undergone out-patient rehabilitation or were assumed to have resumed their pre-morbidity activities of daily living.

In observational studies, non-comparability between the intervention group and the comparison group can distort the estimation of the treatment effect [12,13]. The propensity score is a balancing score that can be used to compare groups that do not systematically differ. This study used PSM at the patient level to compare the baseline characteristics of the two groups, which increased the robustness of the analysis. A generalized estimating equation (GEE) model was used to cluster stroke patients treated by the same physician and to generate propensity scores for predicting the probability of the medical referral system. The covariates included patient demographics (age and gender), clinical attributes (stroke type, hypertension, hyperlipidemia, diabetes mellitus, atrial fibrillation, and previous stroke), quality of medical care (acute care LOS and PAC LOS), and pre-rehabilitation functional status. The caliper matching method (“greedy algorithm”) was used for 1:1 PSM between the inter-hospital transfer group and the intra-hospital transfer group. Thus, 483 patients in the inter-hospital transfer group were compared with an “all participants matched set” of 483 patients in the intra-hospital transfer group (Figure 1). These PAC stroke patients completed the pre-rehabilitation and the 12th week and first year post-rehabilitation assessments.

### 2.3. Functional Status Instruments

The MRS scores of 0, 1, 2, 3, 4, and 5 are interpreted as no symptoms, no significant disability, slight disability, moderate disability, moderately severe disability, and severe disability, respectively [14]. The Barthel Index (BI) score was used to measure functional disability in daily life activities (e.g., eating, grooming, bathing, dressing, walking, transferring, staring, and controlling bladder and bowel) [15]. The score is for the 10-item BI ranges from 0 (totally dependent) to 100 (independent). The Functional Oral Intake Scale (FOIS) was used to assess functional oral intake in stroke patients with dysphagia [16]. The FOIS classifies swallowing function from level 1 (nothing by mouth) to level 7 (total oral diet with no restrictions). The Lawton-Brody Instrumental Activities of Daily Living Scale (IADL) is used to evaluate performance in daily life activities, including making telephone calls, shopping, preparing food, housekeeping, laundering, taking medicine, using transportation, and performing financial activities [17]. In the conventional use of the scale, women are scored in all eight domains, while men are not scored in the domains of preparing food, housekeeping, and laundering. The rationale for excluding these three domains in males is that performance of these tasks is subject to cultural differences in gender roles, which could compromise comparisons of the incidence of disability between men and women [18]. The EuroQoL five-dimensional (EQ-5D) measure is a self-assessment of mobility, self-care, usual activities, pain or discomfort, and anxiety or depression as part of a total health state [19]. The subject is required to score each item from 1 to 3 (no problem, some problem, or extreme problem, respectively). The Berg Balance Scale (BBS) is a scale of functional balance, including static and dynamic balance [20]. Each item on this 14-item scale is rated from 0 (poor balance) to 4 (good balance). The maximum score is 56. The Mini-Mental State Examination (MMSE) is the best-known short screening tool for cognitive impairment [21]. The MMSE tests orientation, attention, memory, language, and visual–spatial skills. The maximum score is 30 points. An MMSE score below an education-adjusted cut-off score indicates cognitive impairment. The Taiwan version of these measures has been validated as a reliable and valid tool for measuring functional status in both clinical practice and research [22].

### 2.4. Statistical Analysis

The unit of analysis in this study was the individual stroke patient. Descriptive statistics were tabulated to depict the stroke patient demographics. For clarification, the values predicted by the regression models were used to illustrate the results, starting from before initiation of PAC until completion of 12 weeks to 1 year of follow up in the two matched study groups. Thus, the GEE models were used to estimate difference-in-differences models used to examine the effectiveness of the medical referral system. For the predicted values, standard errors in differences and standard errors in difference-in-differences were estimated using a bootstrap technique involving 1000 replications, with sample sizes equivalent to that of the original sample [23].

Hierarchical linear regression models were used to examine the roles of the MRS after accounting for demographic characteristics, clinical attributes, quality of care, and pre-rehabilitative function status. Five-step hierarchical linear regression models were used to analyze differences between explanatory factors. In Model 1, explanatory factors included age, gender, stroke type, hypertension, hyperlipidemia, diabetes mellitus, atrial fibrillation, previous stroke, length of stay in acute care, length of stay in post-acute care, and pre-rehabilitation functional status. Model 2 included Model 1 and MRS = 2; Model 3 included Model 1 and MRS = 3; Model 4 included Model 1 and MRS = 4; Model 5 included Model 1, MRS, and medical referral system. In each model, the adjusted *R*-square was estimated while adjusting for covariates. For each model, the variance inflation factor (VIF) was used to assess multicollinearity. No models showed multicollinearity.

Statistical analyses were performed using Stata Statistical Package, version 13.0 (Stata Corp, College Station, TX, USA). All tests were two-sided, and p values less than 0.05 were considered statistically significant.

## 3. Results

Table 1 compares the inter-hospital transfer group and the intra-hospital transfer group before and after PSM. Before PSM, all assessed characteristics significantly differed between the two groups (*p* < 0.05). After PSM, no variables significantly differed between the two groups.

All PAC stroke patients had significantly improved scores for functional measures at the 1-year follow-up survey (*p* < 0.001) (Table 2). When the 12th week post-rehabilitation scores were used as the baseline, functional status showed significant improvements at the first year post-rehabilitation (*p* < 0.001). All subscale scores continued to improve throughout the follow-up period. Additionally, in both groups, functional status scores at the first year post-rehabilitation were significantly higher than the functional status scores at the 12th week post-rehabilitation and the functional status scores at pre-rehabilitation (*p* < 0.001). Throughout the follow-up period, the intra-hospital transfer group also had significantly higher scores for functional status measures compared to the inter-hospital transfer group (*p* < 0.001).

Table 3 shows that Model 1 revealed a significant association between patient demographics and scores for functional status measures at the first year post-rehabilitation during the study period (*p* < 0.05). In Models 2–4, patients with MRS = 2 showed very little functional status improvement after adjustment for patient demographics; however, patients with MRS = 3 and MRS = 4 showed improvements. That is, rehabilitative PAC improved quality of life in patients with moderate disability (MRS = 3) or moderately severe disability (MRS = 4) but not in patients with slight disability (MRS = 2). After adjustment for all relevant influential factors, the intra-hospital transfer group had greater improvements in functional status scores compared to the inter-hospital transfer group.

## 4. Discussion

Our data for the percentage of stroke patients with vascular risk factors were consistent with previous reports. After PSM, the percentages of stroke patients with hypertension, hyperlipidemia, diabetes mellitus, atrial fibrillation in our study were 77%, 42%, 37%, and 10%, respectively. Previous studies of stroke patients have reported hypertension in 63–80%, hyperlipidemia in 40–49%, diabetes mellitus in 34–42%, and atrial fibrillation in 7.3–11% [24,25]. A review of studies performed in Asia found that Taiwan has a higher prevalence of hypertension, diabetes mellitus, and hyperlipidemia compared to Japan, Korea and Singapore [26]. A previous Taiwan study of stroke incidence and recurrence during 2000–2011 also reported that, although the prevalence of diabetes mellitus, hyperlipidemia and atrial fibrillation increased during this period, the rates of primary ischemic stroke and 1-year recurrence of stroke decreased by 9% and 18% respectively [27]. Factors that can have important effects on stroke incidence and recurrence include medication control, early detection of diseases, diet control, body weight control, life style adjustment and exercise.

To achieve a vertically integrated medical system, post-stroke care should be patient-centered, and inter-hospital transfer of stroke patients must be seamless. In Australia, stroke patients treated at hospitals with stroke coordinators had lower LOS and higher quality of evidence-based care compared to those treated at hospitals without stroke coordinators [28]. In the Taiwan national PAC project reported here, the case manager assigned to each stroke patient ensured efficient transfer of the patient to a local hospital. Therefore, the mean LOS for patients in acute stage was much lower in the intra-hospital transfer group compared to the inter-hospital transfer group (13.01 and 24.45 days, respectively). Since minimizing the time from admission to inter-hospital transfer decreases total LOS and total cost, implementation of a case manager and an efficient inter-hospital transfer system are essential for an integrated medical system. Other possible reasons in the inter-hospital transfer group include geographical variations in the distribution of physicians and medical resources, and differences in care quality and expertise among hospitals and individual providers. For stroke patients in acute stage, those treated at teaching hospitals and certified primary stroke centers have a lower mortality rate, greater availability of rehabilitative care, and lower ADL dependence status [29]. A possible reason for the superior outcomes obtained by teaching hospitals and certified primary care centers for stroke is that they tend to have a high volume of stroke patients, and clinicians who treat a high volume of patients tend to achieve high skill levels.

Before PSM, compared to the intra-hospital transfer group, the inter-hospital transfer group in this study had a lower functional status before PAC and more comorbidity (hypertension, hyperlipidemia, atrial fibrillation). Most of the patients in inter-hospital transfer group had been treated at a medical center. Compared to stroke patients treated at medical centers tend to have a higher severity of disability, are more likely to require intubation (e.g., tracheostomy tube, nasogastric tube, and urinary catheter tube), and tend to require more time to stabilize. These differences might explain why the inter-hospital transfer group in our study had a longer mean LOS in acute stage compared to the intra-hospital transfer group. However, further studies are needed to compare the service path and treatment costs in patients with varying severity of stroke and in stroke patients treated at different hospital levels.

After the acute stage, local low-volume rehabilitation facilities can usually provide adequate care [30]. Several studies have reported that, compared to skilled nursing facilities, intensive inpatient rehabilitation facilities achieve higher functional outcomes in PAC for stroke [30,31]. Additionally, duration of hospital stay and in-hospital mortality are related to socioeconomic status [32]. Therefore, increasing socioeconomic inequality over time has markedly increased inequality in stroke survival. Direct non-healthcare costs (including informal care costs, paid care costs and transportation costs) and rehabilitation costs are the largest post-stroke care costs [33,34]. Stroke patients who receive PAC and rehabilitation at local hospitals can reduce the physical, mental and economic burdens on their families. Our study analyzed data obtained from four local southern Taiwan hospitals that had the largest volumes of stroke inpatients in PAC. Therefore, the data obtained in this work are highly representative of hospitals throughout Taiwan and have immediate applications.

In Japan, the rehabilitation is 3 hours, and the average LOS in rehabilitation facilities after discharge from tertiary hospitals is approximately 74.7 days [35]. The Taiwan PAC-CVD program provides the maximum of 12 weeks of services. Reimbursements are similar regardless of the hospital accreditation level (regional or district) and the equipment costs of the hospital. The PAC-CVD project enables stroke patients to access rehabilitation programs that are more intensive (in terms of frequency and duration of treatment) compared to those available under current NHIA provisions. The PAC-CVD project was expected to offer a continuous care model to restore function and reduce disability in stroke patients [8,9]. Most PAC-CVD studies published thus far have been studies of a limited case number in a single hospital [8,9]. In the post-acute stage, the mean LOS in the intra-hospital transfer group and the inter-hospital transfer group was 31.52 and 37.1 days, respectively. Another study of a large population of stroke patients reported a mean PAC stay of only 15.1 days [36]. Notably, the LOS of stroke patients in different countries is related to differences in post-stroke policy. For example, the average LOS for inpatient rehabilitation after stroke is approximately 1 month in Ireland, Switzerland, and Thailand [37]. In contrast, the average LOS inpatient rehabilitation after stroke is only 15.8 days in the United States, which is much shorter than that in western countries (e.g., 32.7 days in Germany, 35.3 days in Canada) [38,39,40]. The main reasons for the short LOS in the United States are the high economic burden of inpatient care for stroke patients and the availability of various well-established PAC rehabilitation facilities (e.g., inpatient rehabilitation hospitals, skilled nursing facilities, home health agency services, long term care hospitals) [4]. According to our multi-center data, the LOS for stroke patients who undergo PAC in Taiwan is similar to that in other countries. Notably, our data indicated that the effectiveness of rehabilitation training during the 12th week to the first year is better than in the first 12 weeks. These data confirm that a continuous rehabilitative PAC program is essential.

A study by Dewilde concluded that MRS level is a major determinant of medical resource use [41]. For this national PAC-CVD project, the MRS level was the main criterion for participation. Initially, only patients with MRS levels of 2 to 4 were eligible for transfer to hospitals with PAC wards. However, some people have proposed excluding patients in MRS level 2 or including patients in MRS level 5. In 2019, the Taiwan NHIA excluded MRS level 2 patients from this national PAC-CVD project. This study found that after adjusting for all relevant influential factors, the quality of life improvement was smaller in patients at MRS level 2 compared to patients at other MRS levels. Therefore, these data support the decision made by the Taiwan NHIA. In terms of maximizing efficiency in the use of limited healthcare resources, the decision to limit inpatient rehabilitation to patients with MRS 3 to 4 was not only justifiable, but necessary.

Although all research questions were adequately and satisfactorily addressed, two limitations are noted. This study only collected data for acute stroke patients for 40 days after stroke onset. Furthermore, this study only analyzed patients treated at four hospitals in south Taiwan. However, the numbers of patients treated in the PAC-programs at these hospitals were among the four highest of all district hospitals in south Taiwan. Further studies are needed to compare a PAC group and a control group in other regions of Taiwan and under current NHI regulations. Additionally, the two groups in this study were matched for demographic characteristics, clinical attributes, quality of care, and pre-rehabilitative functional status. Future studies could consider the use of inverse probability weighting rather than PSM.

## 5. Conclusions

In conclusion, this study showed that rehabilitative PAC improved outcomes of stroke rehabilitation. To achieve a vertically integrated medical system for stroke rehabilitation, the key requirements are improving the PAC ward qualifications of local hospitals, accelerating inter-hospital transfer, and ensuring a sufficient duration of intensive rehabilitative PAC. Early rehabilitation is important for successful restoration of health, confidence, and self-care ability in these patients.

## Figures and Tables

**Figure 1 jcm-08-01233-f001:**
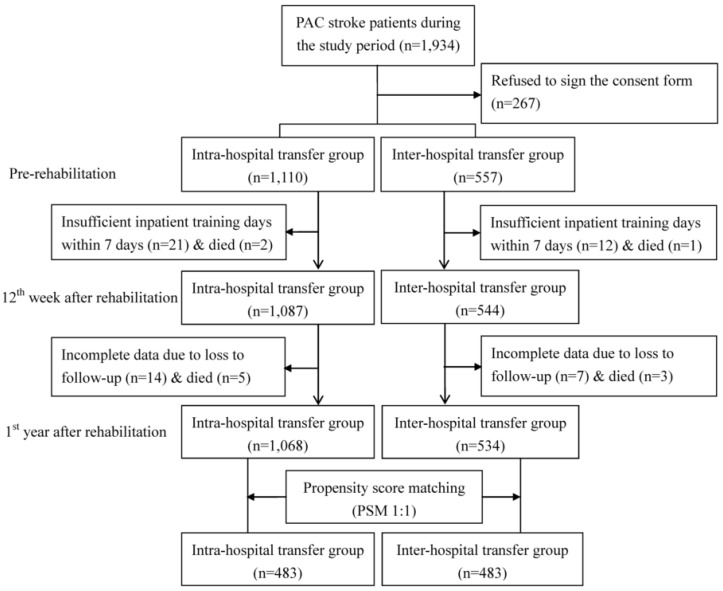
Flow chart of recruitment and study procedure.

**Table 1 jcm-08-01233-t001:** Stroke patient characteristics *.

		Before Propensity Score Matching			After Propensity Score Matching		
Variables		Intra-Hospital Transfer Group (*n* = 1068)	Inter-Hospital Transfer Group (*n* = 534)	*p* Value	Intra-Hospital Transfer Group (*n* = 483)	Inter-Hospital Transfer Group (*n* = 483)	*p* Value
Demographics							
Age, years *		65.67 ± 12.38	63.96 ± 13.50	0.024	63.61 ± 13.10	63.96 ± 13.24	0.834
Gender	Female	450(42.1%)	186(34.8%)	0.014	171(35.4%)	169(35.0%)	0.784
	Male	618(57.9%)	348(65.2%)		312(64.6%)	314(65.0%)	
Clinical Attributes							
Stroke type	Ischemic	942(88.2%)	396(74.2%)	<0.001	363(75.0%)	360(74.5%)	0.880
	Hemorrhagic	126(11.8%)	138(25.8%)		120(25.0%)	123(25.5%)	
Hypertension	Yes	698(65.4%)	405(75.8%)	<0.001	372(77.0%)	367(76.0%)	0.221
Hyperlipidemia	Yes	332(31.1%)	227(42.5%)	<0.001	202(41.8%)	206(42.6%)	0.507
Diabetes mellitus	Yes	419(39.2)	200(37.5%)	0.570	182(37.7%)	181(37.5%)	0.990
Atrial fibrillation	Yes	67(6.3%)	55(10.3%)	0.013	47(9.7%)	49(10.1%)	0.887
Previous stroke	Yes	173(16.2%)	98(18.4%)	0.803	87(18.0%)	89(18.4%)	0.879
Quality of Medical Care							
Acute care LOS, days *		13.01 ± 27.83	24.45 ± 34.61	<0.001	23.75 ± 11.84	24.50 ± 11.56	0.356
PAC LOS, days *		31.52 ± 17.75	37.1 ± 12.59	<0.001	35.75 ± 12.34	36.50 ± 11.88	0.506
Pre-rehabilitation functional status							
BI *		41.91 ± 23.10	34.67 ± 23.48	<0.001	35.75 ± 20.11	34.00 ± 18.21	0.269
FOIS *		5.95 ± 3.04	5.38 ± 2.25	<0.001	5.53 ± 2.75	5.14 ± 2.84	0.927
EQ5D *		10.67 ± 1.86	10.40 ±1.78	0.015	10.80 ± 1.82	10.93 ± 2.05	0.261
IADL *		1.41 ± 1.20	1.15 ± 1.12	<0.001	1.32 ± 1.14	1.27 ± 1.05	0.694
BBS *		15.30 ± 14.99	16.91 ± 17.27	0.097	14.00 ± 17.26	15.50 ± 17.71	0.972
MMSE *		20.15 ± 7.90	18.50 ± 9.66	0.001	20.75 ± 11.15	19.75 ± 10.47	0.908

Note: LOS, length of stay; PAC, post-acute care; QOL, quality of life; BI, Barthel Index; FOIS, Functional Oral Intake Scale; EQ5D, EuroQoL five-dimensional; IADL, Instrumental Activities of Daily Living Scale; BBS, Berg Balance Scale; MMSE, Mini-Mental State Examination. * Values are expressed as mean ± standard deviation or *n* (%).

**Table 2 jcm-08-01233-t002:** Comparison of functional status between intra-hospital transfer group and inter-hospital transfer group, before and after rehabilitation (*n* = 1195).

Functional Status Measure	Group	Before Rehabilitation (T1)	12th Week After Rehabilitation (T2)	Difference ^†^	*p* Value	First Year After Rehabilitation (T3)	Difference ^†^	*p* Value
		Mean ± SD				Mean ± SD		
BI	Intra-hospital transfer	41.91 ± 23.10	51.50 ± 24.10	9.59	<0.001	68.84 ± 26.49	17.34	<0.001
	Inter-hospital transfer	34.67 ± 23.48	42.28 ± 25.96	7.61	<0.001	54.29 ± 27.20	12.01	0.002
	Difference ^‡^	7.24	9.22	1.98	<0.001	14.55	5.33	<0.001
FOIS	Intra-hospital transfer	5.95 ± 3.04	5.98 ± 1.82	0.03	0.037	6.41 ± 1.32	0.43	0.006
	Inter-hospital transfer	5.38 ± 2.25	5.68 ± 1.89	0.30	0.002	6.26 ± 1.47	0.58	0.044
	Difference ^‡^	0.57	0.30	−0.27	<0.001	0.15	−0.15	<0.001
EQ5D	Intra-hospital transfer	10.67 ± 1.86	9.81 ± 1.65	−0.86	<0.001	8.15 ± 2.23	−1.66	<0.001
	Inter-hospital transfer	10.40 ± 1.79	10.02 ± 1.70	−0.38	0.004	9.19 ± 1.88	−0.83	0.001
	Difference ^‡^	0.27	−0.21	−0.48	<0.001	−1.04	−0.83	<0.001
IADL	Intra-hospital transfer	1.41 ± 1.20	1.84 ± 1.32	0.43	<0.001	2.87 ± 1.75	1.03	<0.001
	Inter-hospital group	1.15 ± 1.13	1.36 ± 1.31	0.21	0.054	1.86 ± 1.57	0.5	0.023
	Difference ^‡^	0.26	0.48	0.22	<0.001	1.01	0.53	<0.001
BBS	Intra-hospital transfer	15.30 ± 14.99	26.32 ± 17.56	11.02	<0.001	34.58 ± 17.79	8.26	<0.001
	Inter-hospital transfer	16.91 ± 17.27	24.40 ± 19.25	7.49	<0.001	29.91 ± 19.35	5.51	0.022
	Difference ^‡^	−1.61	1.92	3.53	<0.001	4.67	2.75	<0.001
MMSE	Intra-hospital transfer	20.15 ± 7.90	21.62 ± 7.79	1.47	0.020	22.73 ± 7.39	1.11	0.078
	Inter-hospital transfer	18.50 ± 9.66	19.64 ± 10.80	1.14	0.090	21.25 ± 9.41	1.61	0.260
	Difference ^‡^	1.65	1.98	0.33	<0.001	1.48	−0.5	<0.001

Note: BI, Barthel Index; FOIS, Functional Oral Intake Scale; EQ5D, EuroQoL five-dimensional; IADL, Instrumental Activities of Daily Living Scale; BBS, Berg Balance Scale; MMSE, Mini-mental State Examination. ^†^ Difference indicates mean score for functional status at the 12th week after rehabilitation, mean score for functional status before rehabilitation, or mean score for functional status at the first year after rehabilitation. ^‡^ Difference indicates mean score for functional status in intra-hospital transfer group, or mean score for functional status in inter-hospital transfer group at each time point.

**Table 3 jcm-08-01233-t003:** Change in coefficient of multiple correlations associated with addition of subsequent variables to the model *.

Model	BI		FOIS		EQ5D		IADL		BBS		MMSE	
	*R* ^2^	*R*^2^ Change	*R* ^2^	*R*^2^ Change	*R* ^2^	*R*^2^ Change	*R* ^2^	*R*^2^ Change	*R* ^2^	*R*^2^ Change	*R* ^2^	*R*^2^ Change
1st level	0.64	-	0.28	-	0.38	-	0.51	-	0.48	-	0.79	-
2nd level	0.64	0.00	0.28	0.00	0.39	0.01	0.52	0.01	0.48	0.00	0.79	0.00
3rd level	0.64	0.00	0.29	0.01	0.44	0.06	0.55	0.04	0.51	0.03	0.79	0.00
4th level	0.64	0.00	0.29	0.01	0.44	0.06	0.55	0.04	0.51	0.03	0.79	0.00
5th level	0.67	0.03	0.29	0.01	0.49	0.11	0.58	0.07	0.53	0.05	0.79	0.00

Note: BI, Barthel Index; FOIS, Functional Oral Intake Scale; EQ5D, EuroQoL five-dimensional; IADL, Instrumental Activities of Daily Living Scale; BBS, Berg Balance Scale; MMSE, Mini-mental State Examination. * Model 1 included age, gender, stroke type, hypertension, hyperlipidemia, diabetes mellitus, atrial fibrillation, previous stroke, length of stay in acute care, length of stay in post-acute care, and pre-rehabilitation functional status; Model 2 included Model 1 and Modified Rankin Scale (MRS) = 2; Model 3 included Model 1 and MRS = 3; Model 4 included Model 1 and MRS = 4; Model 5 included Model 1, MRS, and medical referral system.

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
