# Peer review of "Multidiscipline Stroke Post-Acute Care Transfer System: Propensity-Score-Based Comparison of Functional Status"

_jcm, 2019, doi:10.3390/jcm8081233_

Round 1
Reviewer 1 Report
Abstract:
This sentence “Few studies have investigated how the characteristics of stroke inpatients after post-acute care (PAC) rehabilitation” is incomplete. How what?
Lines 33-34 do not read well. I would suggest splitting this long sentence into two and then clearly define the exposure of interest and the outcome
Line 39: remove “also”
Lines 41-42: effect on what? Again reading the abstract, I am confused what the primary exposure of interest is and what is the outcome. These must be very clearly stated.
“Rehabilitative PAC was an effective training for the stroke” What kind of training for stroke? This does not seem to make sense.
……..most essential issues” I don’t think issues is the word the authors want to use.
Please make sure that this study is proof read by a native English speaker. In its current form it is difficult to understand what the authors are trying to say. I would not point out such issues further. It seems that abstract section was not proof read before submission because the sections after this read pretty well mostly.
Line 53: The first sentence seems incomplete.
The introduction section while is framed well, is very limited. The authors do not provide sufficient evidence of what’s been done and what are the data gaps exactly. Just mentioning that there are fewer papers is not enough.
The abstract section needs to include the study objectives as described in the introduction section.
Lines 84-85, please cite the relevant references for this.
What is rationale for taking “Modified Rankin Scale (MRS) level 2 to 4, where MRS scores of 0, 1, 2, 3, 4 and 5 are interpreted as no symptoms”? Why were 1 and 5 not included? Are all of these no symptoms (does not make sense)?
“Potential selection bias was further minimized” How do the sections earlier minimize selection bias?
There are too many abbreviations in this study, please try and limit these.
A lot of references are missing throughout. For example, reference the literature for line 94.
Consider sensitivity analyses with those excluded. Were these different from those enrolled? This will help partially assess any potential selection bias.
After PSM was done (Figure 1), seems like there were quite a few patients for which match was not found. This would be fine if those excluded are truly different from the remaining sample but if not, this could introduce bias. Note that a large proportion of the “intra” group were excluded. Please consider sensitivity analyses for this and also acknowledge this in the limitations section. Inverse probability weights could have been considered instead of PSM.
Limes 140-141: There could be multiple testing related issues
Table 2: I am confused; dependent variables were not continuous (for FOIS is limited between levels 1 and 7) but the table seems to suggest that linear regression model was used. Please clearly specify what “link” and “distribution” were specified for GEE models for each of the outcomes.
Table 3: Not sure what this table adds. The R2 increasing as more variables are added. What is the point here? Also is this adjusted R2? I do not think this table is needed
Lines 174-175: Please provide the estimates you found and what previous research found. Just mentioning that they were similar is not enough.
Lines 176-177: These are not relevant to this paper. These were not the exposures of interest specifically but mentioning them here specifically creates confusion.
Author Response
Reviewer 1
Comments and Suggestions for Authors
Abstract:
This sentence “Few studies have investigated how the characteristics of stroke inpatients after post-acute care (PAC) rehabilitation” is incomplete. How what?
Ans:
Thank you for your comment. We added the statement in the sentence to make it clear and the word “how” has been deleted from the sentence.
Lines 33-34 do not read well. I would suggest splitting this long sentence into two and then clearly define the exposure of interest and the outcome
Ans:
As advised, the sentence has been split into two shorter sentences, and the exposure of interest and the outcome have been defined. Thank you.
Line 39: remove “also”
Ans:
The word “also” has been deleted. Thank you.
Lines 41-42: effect on what? Again reading the abstract, I am confused what the primary exposure of interest is and what is the outcome. These must be very clearly stated.
Ans:
The revised abstract clarifies the exposure of interest and outcome investigated in this study. Thank you.
“Rehabilitative PAC was an effective training for the stroke” What kind of training for stroke? This does not seem to make sense.
Ans:
This statement has been deleted from the Abstract. Thank you.
……..most essential issues” I don’t think issues is the word the authors want to use.
Ans:
The word “issues” has been changed to “requirements.” Thank you.
Please make sure that this study is proof read by a native English speaker. In its current form it is difficult to understand what the authors are trying to say. I would not point out such issues further. It seems that abstract section was not proof read before submission because the sections after this read pretty well mostly.
Ans:
Thank you. The resubmitted abstract and manuscript have been proofread by an experienced technical editor.
Line 53: The first sentence seems incomplete.
Ans:
The sentence has been rewritten (lines 2-5, page 3 in revised version). Thank you.
The introduction section while is framed well, is very limited. The authors do not provide sufficient evidence of what’s been done and what are the data gaps exactly. Just mentioning that there are fewer papers is not enough.
Ans:
As advised, the revised Introduction section mentions that recent studies have analyzed a limited number of patients in a single hospital, have used longitudinal data exceeding 1 year, and have applied propensity score matching (PSM) in a natural experimental design to examine the longitudinal impacts of PAC program on functional status and discusses the gaps in the literature (lines 2-7, page 4). Thank you.
The abstract section needs to include the study objectives as described in the introduction section.
Ans:
As advised, the revised Abstract states the objectives of the study. Thank you.
Lines 84-85, please cite the relevant references for this.
Ans:
Two relevant references are cited. Thank you.
What is rationale for taking “Modified Rankin Scale (MRS) level 2 to 4, where MRS scores of 0, 1, 2, 3, 4 and 5 are interpreted as no symptoms”? Why were 1 and 5 not included? Are all of these no symptoms (does not make sense)?
Ans:
The resubmitted version clarifies that MRS levels 1 and 5 were excluded because patients who did not have major disability (MRS 0-1) were assumed to have undergone out-patient rehabilitation or were assumed to have resumed their premorbidity activities of daily living (MRS 5) (lines 14-19, page 5). Thank you.
“Potential selection bias was further minimized” How do the sections earlier minimize selection bias?
Ans:
Following for your comments above, we revised these statements to make it clear (lines 3-8, page 6). Thank you.
There are too many abbreviations in this study, please try and limit these.
Ans:
Following for your comment above, abbreviations were defined in parentheses the first time they appear in the main text and used consistently thereafter. Thank you.
A lot of references are missing throughout. For example, reference the literature for line 94.
Ans:
The manuscript has been reviewed to ensure that sources are cited as needed. Thank you.
Consider sensitivity analyses with those excluded. Were these different from those enrolled? This will help partially assess any potential selection bias.
Ans:
Following for your comments above, we revised these statements to make it clear (lines 3-8, page 6). Thank you.
After PSM was done (Figure 1), seems like there were quite a few patients for which match was not found. This would be fine if those excluded are truly different from the remaining sample but if not, this could introduce bias. Note that a large proportion of the “intra” group were excluded. Please consider sensitivity analyses for this and also acknowledge this in the limitations section. Inverse probability weights could have been considered instead of PSM.
Ans:
The revised Study Design and Sample section (lines 3-8, page 6 & lines 14-17, page 6) and Limitations section (lines 14-17, page 17) clarify that a large proportion of the “intra” group were excluded. Thank you.
Limes 140-141: There could be multiple testing related issues
Ans:
To address the issue of multiple testing, the resubmitted version describes the use of five-step hierarchical linear regression models to analyze differences between explanatory factors (lines 8-19, page 9). Thank you.
Table 2: I am confused; dependent variables were not continuous (for FOIS is limited between levels 1 and 7) but the table seems to suggest that linear regression model was used. Please clearly specify what “link” and “distribution” were specified for GEE models for each of the outcomes.
Ans:
We used FOIS for continuous variables as described in the literature (Yi YG, et al. Dysphagia. 2019;34:201-209; Wilmskoetter J, et al. J Stroke Cerebrovasc Dis. 2019;28:1421-1430; Sakai K, et al. Dysphagia. 2017;32:241-249; Wang CY, et al. Int J Qual Health Care. 2017;29:779-784). Thank you.
Table 3: Not sure what this table adds. The R2 increasing as more variables are added. What is the point here? Also is this adjusted R2? I do not think this table is needed
Ans:
The purpose of the table is to examine the roles of the MRS after accounting for demographic characteristics, clinical attributes, quality of care and pre-rehabilitative function status. We added the statements in the Statistical Analysis section (lines 8-19, page 9). Thank you.
Lines 174-175: Please provide the estimates you found and what previous research found. Just mentioning that they were similar is not enough.
Ans:
As advised, the revised Discussion section compares our data with that in previous reports (lines 16-19, page 11 and lines 1-2, page 12). Thank you.
Lines 176-177: These are not relevant to this paper. These were not the exposures of interest specifically but mentioning them here specifically creates confusion.
Ans:
Thank you for your comment. The two sentences have been deleted.
Reviewer 2 Report
It would be helpful for an unfamiliar reader to know more details about the PAC-CVD transfer policy and how this determines inter-hospital vs. intra-hospital transfers. There are multiple, relatively minor grammatical or syntax errors in English that some focused proofreading can fix. For the IADL scale, it is not clear whether the authors used the gender breakdown where there is a maximum score of 5 for men and 8 for women. If used, then results should be stratified by gender or the authors should comment in greater detail on the methods used for this analysis. Please comment in the discussion about whether patients with Modified Rankin Scale are excluded from PAC in other countries, to your knowledge, in the dicussion section.
Author Response
Reviewer 2
Comments and Suggestions for Authors It would be helpful for an unfamiliar reader to know more details about the PAC-CVD transfer policy and how this determines inter-hospital vs. intra-hospital transfers. There are multiple, relatively minor grammatical or syntax errors in English that some focused proofreading can fix. For the IADL scale, it is not clear whether the authors used the gender breakdown where there is a maximum score of 5 for men and 8 for women. If used, then results should be stratified by gender or the authors should comment in greater detail on the methods used for this analysis. Please comment in the discussion about whether patients with Modified Rankin Scale are excluded from PAC in other countries, to your knowledge, in the dicussion section.
Ans:
Firstly, the Study Design and Sample section clarifies the inter-hospital and the intra-hospital transfer groups (lines 14-16, page 6); secondly, all revisions have been reviewed by a professional technical editor; thirdly, the revised Functional Statement Instruments section explains the rationale for excluding these three domains in males is that performance of these tasks is subject to cultural differences in gender roles, which could compromise comparisons of the incidence of disability between men and women (lines 17-19, page 7); finally, hierarchical linear regression models were used to examine the role of MRS after accounting for demographic characteristics, clinical characteristics, quality of care and pre-rehabilitative function status (lines 8-19, page 9). Additionally, for each model, the adjusted R-square was estimated while adjusting for covariates. The Discussion section has been revised accordingly (lines 1-6, page 17). Thank you.